# Flux analysis of cholesterol biosynthesis in vivo reveals multiple tissue and cell-type specific pathways

Matthew A Mitsche[1], Jeffrey G McDonald[1], Helen H Hobbs[1,2]*,
Jonathan C Cohen[3]*

[1]Department of Molecular Genetics, University of Texas Southwestern Medical Center, Dallas, United States; [2]Howard Hughes Medical Institute, University of Texas Southwestern Medical Center, Dallas, United States; [3]Center for Human Nutrition, University of Texas Southwestern Medical Center, Dallas, United States

**Abstract** Two parallel pathways produce cholesterol: the Bloch and Kandutsch-Russell pathways. Here we used stable isotope labeling and isotopomer analysis to trace sterol flux through the two pathways in mice. Surprisingly, no tissue used the canonical K–R pathway. Rather, a hybrid pathway was identified that we call the modified K–R (MK–R) pathway. Proportional flux through the Bloch pathway varied from 8% in preputial gland to 97% in testes, and the tissue-specificity observed in vivo was retained in cultured cells. The distribution of sterol isotopomers in plasma mirrored that of liver. Sterol depletion in cultured cells increased flux through the Bloch pathway, whereas overexpression of 24-dehydrocholesterol reductase (DHCR24) enhanced usage of the MK–R pathway. Thus, relative use of the Bloch and MK–R pathways is highly variable, tissue-specific, flux dependent, and epigenetically fixed. Maintenance of two interdigitated pathways permits production of diverse bioactive sterols that can be regulated independently of cholesterol.

*For correspondence: helen. hobbs@utsouthwestern.edu (HHH); jonathan.cohen@ utsouthwestern.edu (JCC)

## Introduction

Cholesterol is an essential structural component of vertebrate cell membranes (*Maxfield and Tabas, 2005*) and a precursor of vital end products such as bile acids (*Russell, 2009*) and steroid hormones (*Sih and Whitlock, 1968*). First identified as a crystalline component of gallstones more than 200 years ago, cholesterol consists of a rigid, planar tetracyclic nucleus and a flexible, iso-octyl side-chain at carbon 17 (*Nes, 2011*). The molecule is synthesized entirely from acetate through a complex series of over 30 enzymatic reactions that are clustered into four major processes: condensation of acetate to isoprene, polymerization of isoprene to squalene, cyclization of squalene to lanosterol, and finally the conversion of lanosterol to cholesterol (*Figure 1*).

Two intersecting pathways have been described for biosynthesis of cholesterol from lanosterol. The two pathways use the same catalytic steps and are distinguished by the stage at which the double bond at C24 in the side chain is reduced. *Bloch et al. (1965)* proposed that reduction of the double bond in the side chain (Δ24) is the last reaction in the pathway (*Figure 1*, black arrows). Thus, in the Bloch pathway, cholesterol synthesis proceeds via a series of side-chain unsaturated intermediates to desmosterol, which is reduced by DHCR24 to cholesterol. Subsequently, Kandutsch and Russell (*Kandutsch and Russell, 1960a*, *1960b*) reported that the preputial glands of mice synthesized dihydrolanosterol and other side-chain saturated intermediates that were different from those in the Bloch pathway. They proposed an alternative pathway, the Kandutsch–Russell (K–R) pathway, in which the Δ24 bond of lanosterol is reduced and the conversion of dihydrolanosterol to cholesterol proceeds via 7-dehydrocholesterol (*Figure 1*, red arrows) using the same enzymes as

**eLife digest** Cholesterol is important for animals, both as an essential component of the membrane that surrounds cells and as a building block to make hormones and other biologically important molecules. However, cells limit how much cholesterol they make because an excess of this fatty molecule can cause serious health problems, including heart disease and stroke.

Cholesterol is made via a complex process that involves more than 30 different steps, which can be organized into two biochemical pathways (named the Bloch pathway and the Kandutsch–Russell pathway). The enzymes that carry out the steps in these pathways have been characterized in detail. Less is known about which of the two pathways is actually used in different cells and tissues, or how much cholesterol each pathway produces. This is partly because it is difficult to distinguish between the closely related intermediate molecules that are formed in each pathway.

Mitsche et al. have now used mass spectrometry and isotope labeling techniques to analyze the relative contributions of the two cholesterol-making pathways in both cells grown in the laboratory and in mice. The experiments show that many cells use the Bloch pathway. However, no cells were found to use the Kandutsch–Russell pathway as it was originally described. Rather, some of the cells used a hybrid pathway where the production of cholesterol was started using the Bloch pathway and then after a certain number of steps, the process switched to using part of the Kandutsch–Russell pathway. Mitsche et al. referred to this mixed system as the 'modified Kandutsch–Russell pathway'.

Mitsche et al. next examined the flow of molecules through these two pathways in different tissues and observed that the Bloch pathway is exclusively used in the testes and adrenal glands, which produce high levels of cholesterol. In contrast, the skin and brain use the modified Kandutsch–Russell pathway. In some tissues, a fraction of the building blocks that can be used to make cholesterol were instead diverted to make other products. This suggests that animals have maintained the two pathways over the course of evolution to enable them to generate a variety of products, which can be used to carry out different biological processes. One challenge following this work will be to use the newly developed methods to analyze other complex biochemical pathways.

the Bloch pathway. Later studies showed the presence of saturated side chain intermediates in brain, skin, and eventually in all tissues examined (*Schroepfer, 1982*). The catalytic mechanisms and regulation of the enzymes that catalyze the Bloch and K–R pathways have been extensively investigated but much less is known about their relative use and physiological significance. Cholesterol biosynthesis is usually measured using radioisotope methods (*Dietschy and Spady, 1984*) that are highly sensitive but do not provide information about the turnover of intermediate sterols in the biosynthetic pathway. Flux through the Bloch and K–R pathways has not been systematically studied in cultured cells or in vivo. Therefore, the relative use of the two pathways and their responses to changes in cholesterol availability are not known. The reason why both pathways have been maintained is also not known. The biosynthetic intermediates of the two pathways have powerful, but distinct, effects on cholesterol homeostasis (*Yang et al., 2006*; *Lange et al., 2008*), fatty acid synthesis (*Spann et al., 2012*), and inflammation (*Spann et al., 2012*). Varying the concentrations of specific cholesterol biosynthetic intermediates by differential use of the Bloch and K–R pathways may thus contribute to regulation of diverse cellular processes.

The analytical challenges associated with distinguishing the multiple closely related sterol intermediates in each pathway, and interpreting the complex patterns of isotope enrichment generated by in vivo labeling have been major obstacles to in vivo studies of the Bloch and K–R pathways. *Kelleher and Masterson (1992)* demonstrated that stable isotope methods and isotopomer spectral analysis (ISA) could be used to measure the flux of cholesterol biosynthetic intermediates such as lathosterol in cultured cells and in living animals using gas chromatography–MS (*Lindenthal et al., 2002*). More recently, *McDonald et al. (2012)* developed a liquid chromatography tandem mass spectrometry (LC-MS/MS) method to determine steady-state concentrations of most of post squalene cholesterol biosynthetic intermediates of the Bloch and K–R pathways in biological fluids. Here we combined these approaches to examine the flux of substrate though the Bloch and K–R pathways in cultured cells and in vivo in the tissues of mice.

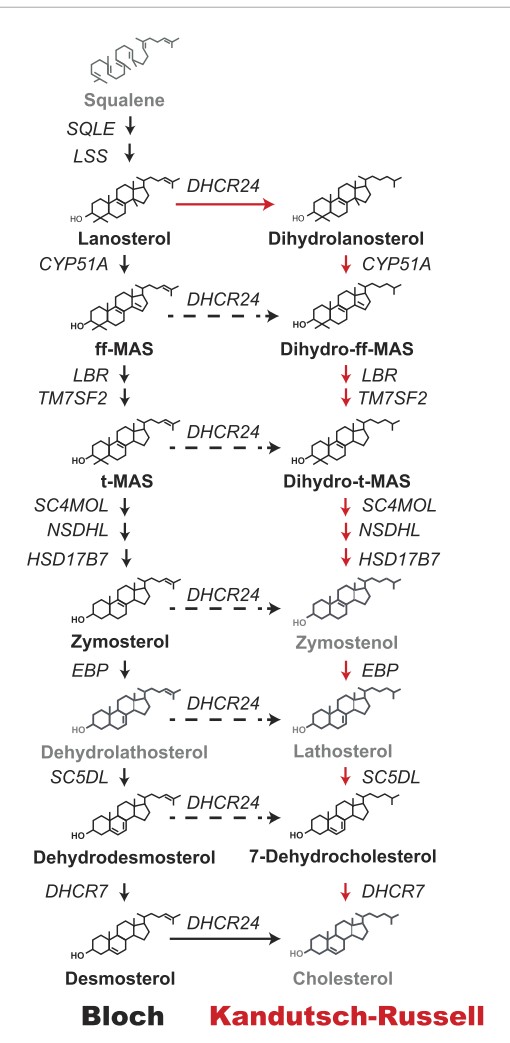

**Figure 1**. Schematic representation of the Bloch and Kandutsch–Russell (K–R) pathways for the enzymatic conversion of squalene to cholesterol. The Bloch pathway, indicated by solid black arrows, is shown on the left. The Kandutsch–Russell pathway, indicated by red arrows, is shown on the right. Additional potential sites of crossover from the Bloch to K–R pathway are indicated by broken arrows. Sterol intermediates that were not measured using deuterium water labeling are shown in gray.

# Results

## Applying stable isotopes to measure sterol biosynthesis rates

As a first step towards characterizing the flux of precursor sterols through the cholesterol biosynthetic pathway, we used LC-MS/MS and deuterium oxide ($D_2O$) labeling to analyze the turnover of a representative sterol, lanosterol, in immortalized human skin fibroblasts (SV-589 cells). The isotopomer distributions of lanosterol before and 24 hr after addition of 5% $D_2O$ to the medium are shown in *Figure 2A*. In unlabeled cells, the isotopomer spectrum matched the predicted distribution based on a 1.1% natural abundance of $^{13}C$ and a 0.015% natural abundance of deuterium (*Figure 2A*, top, left panel). Lanosterol molecules containing exclusively $^{12}C$, $^{16}O$ and $^{1}H$ isotopes had an *m/z* peak of 409 Da (M = 0) and comprised 72% of the total. Lanosterol molecules containing a single extra neutron (i.e., one atom of $^{13}C$ or $^{2}H$) had an *m/z* peak of 410 Da (M = 1) comprised 24% of the total while those containing two extra neutrons (M = 2) comprised 4%.

Cells grown in the presence of $D_2O$ incorporate deuterium into newly synthesized sterols, primarily via deuterated NADPH, causing a decrease in the abundance of M0 isotopomers and an increase in the proportion of heavier molecules. After cells were grown in 5% $D_2O$ long enough for the lanosterol pool to be replaced (in this case, for 24 hr), only 25% of the lanosterol had not incorporated any deuterium atoms (M = 0). Over 75% of the lanosterol molecules had incorporated at least one heavy atom (M1 + M2 + M3) (*Figure 2A*, top, right panel). The largest fraction of lanosterol isotopomers (37%) contained a single deuterium or $^{13}C$ atom (M = 1). The shift in isotopomer distribution over time was used to infer the incorporation of deuterium, which was then used to measure the rate of lanosterol synthesis as described in the 'Materials and methods' and reviewed in *Figure 2—figure supplement 1*.

Next, we examined the effect of 25-hydroxycholesterol (25-OH Chol), a potent suppressor of cholesterol biosynthesis (*Kandutsch et al., 1977*), on lanosterol turnover in cultured fibroblasts (*Figure 2A*, bottom, left panel). Cells were grown for 16 hr in the presence or absence of 25-OH Chol (1 µg/ml) prior to addition of 5% $D_2O$ to the medium. The rate of lanosterol biosynthesis was determined by sampling cells at the indicated time points. At each time point the fraction of newly synthesized lanosterol molecules (termed 'g') was determined from the isotopomer spectrum.

The relationship between time and g was fitted to a first-order kinetic model to determine the rate constant (k), which was multiplied by the lanosterol concentration to determine the rate of synthesis (ng/hr/µg protein). The addition of 25-OH Chol to the medium decreased the rate of lanosterol biosynthesis by 90% (*Figure 2A*, bottom).

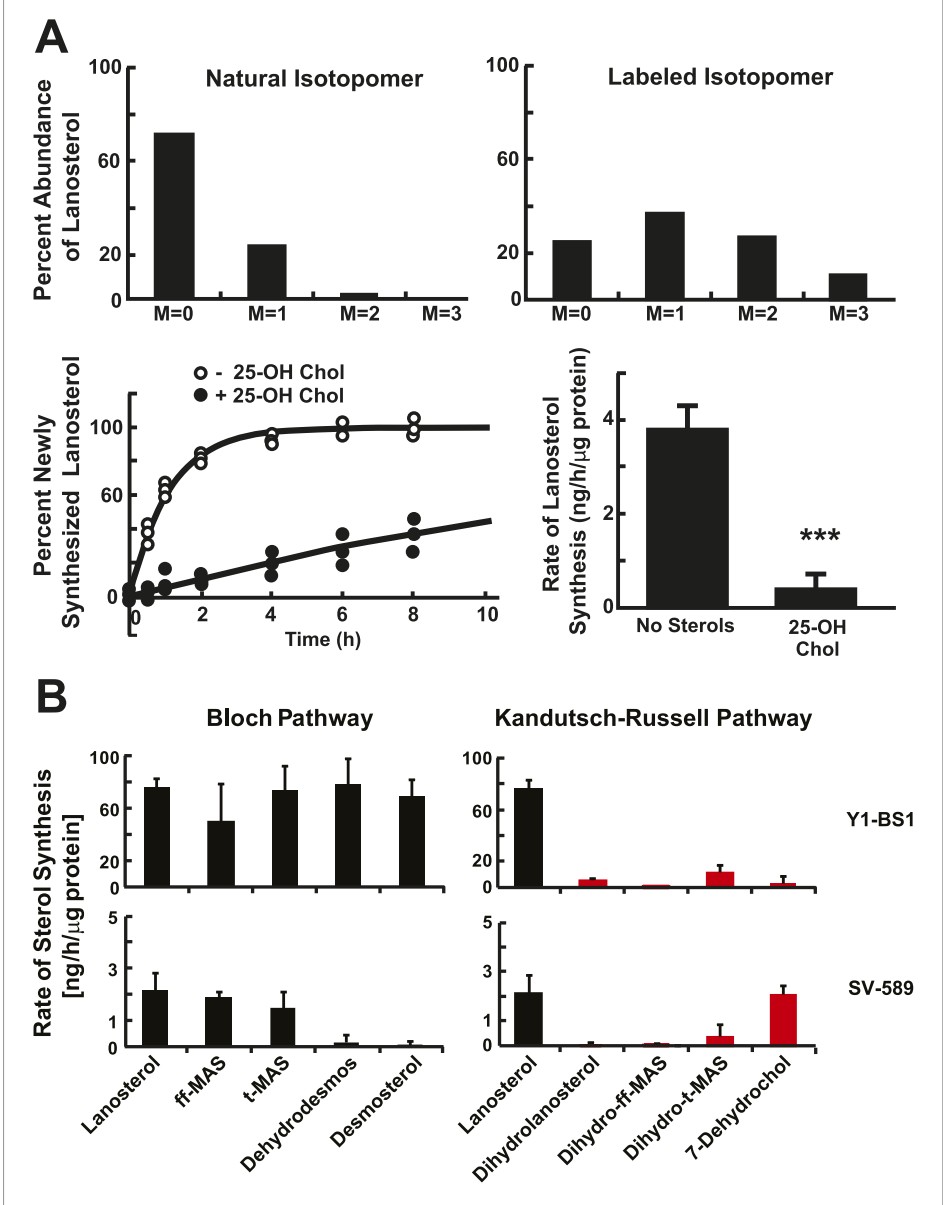

**Figure 2**. Sterol biosynthesis in cultured cells. (**A**) Deuterated water (D$_2$O) labeling of lanosterol in cultured cells. (Top) Isotopomer spectrum of lanosterol in SV-589 cells grown for 24 hr in the absence (left) or presence (right) of 5% D$_2$O added to the medium. (Bottom) Turnover of lanosterol in SV-589 cells. Cells were grown in NCLPPS (open circles) or in NCLPPS plus 25-hydroxycholesterol (1 µg/ml) (closed circles) to ~60% confluence. After 16 hr , the medium was supplemented with 5% D$_2$O and cells were harvested at 0, 0.5, 1, 2, 4, 6, 8, 12, and 24 hr (last two points not shown, but used for modeling). Sterols were analyzed by LC-MS/MS and the fraction of lanosterol that was newly synthesized was determined using isotopomer analysis (IA) (see *Figure 2—figure supplement 1*) and the results were fit to a first-order kinetic model (solid lines) as described in the 'Materials and methods'. The rates of synthesis of lanosterol in the presence or absence of sterols were calculated by multiplying the first-order rate constant by the concentration of lanosterol. Standard deviations are reported based on four independent replicates. (**B**) Biosynthetic rates of intermediary sterols in cultured cells. The rate of synthesis of intermediary sterols in the Bloch (left) and K–R (right) pathways was measured using D$_2$O labeling. Cells were plated at a density of 500,000/60 mm dish and grown to ~60% confluence. D$_2$O was then added to the medium to a final concentration of 5% (vol/vol). Cells were harvested at 0, 0.5, 1, 2, 4, 6, 8, 12, and 24 hr, and lipids were extracted using methanol-dichloromethane. Sterols were analyzed by LC-MS/MS and the fraction of each sterol that was newly synthesized was determined using IA and the results were fit to a first-order kinetic model as described in the 'Materials and methods'. Y1-BS1 cells, a mouse adrenal cell line (Top) were grown in 15%horseserum. SV-589 cells, an immortalized human skin fibroblast

*Figure 2. continued on next page*

*Figure 2. Continued*

line (bottom), were grown in either 10% FCS. Means and standard deviations are reported based on four independent replicate experiments in each cell line. ***p < 0.001.

The following figure supplement is available for figure 2:

**Figure supplement 1**. Schematic representation of IA adapted from *Kelleher and Masterson (1992)*.

## Relative utilization of the Bloch and K–R pathways in cultured adrenal cells and fibroblasts

To determine the relative utilization of the Bloch and K–R pathways in various cell types, we measured and compared the rates of deuterium incorporation from $D_2O$ into post-squalene cholesterol biosynthetic intermediates in cultured mouse adrenal cells (Y1-BS1 cells) (*Watt and Schimmer, 1981*) and transformed human fibroblasts (SV-589 cells) (*Yamamoto et al., 1984*) (*Figure 2B*).

Deuterium was incorporated almost exclusively into Bloch pathway intermediates in Y1BS1 cells, in which the rates of incorporation were similar for lanosterol, ff-MAS, t-MAS, dehydrodesmosterol, and desmosterol (*Figure 2B*, top left panel). Little turnover of K–R intermediates was detected in these cells (*Figure 2B*, top right panel).

In SV-589 fibroblasts (*Figure 2B*, bottom panel), lanosterol was quantitatively converted to ff-MAS and t-MAS (Bloch pathway) with almost no detectable incorporation into the corresponding K–R intermediates (dihydro-ff-MAS and dihydro-t-MAS). Incorporation into the downstream Bloch intermediates dehydrodesmosterol and desmosterol was minimal, but robust labeling of 7-dehydrocholesterol was observed, indicating a crossover from the Bloch to the K–R pathway between t-MAS and dehydrodesmosterol (*Figure 1*). The methylated biosynthetic intermediates between lanosterol and 7-dehydrocholesterol in the K–R pathway (i.e., dihydrolanosterol, dihydro-ff-MAS, and dihydro-t-MAS) did not turnover at comparable rates to either lanosterol or 7-dehydrocholesterol, suggesting that the classical K–R pathway was not used to synthesize cholesterol in these cells. Instead, the cells used a hybrid pathway that we will refer to as the modified Kandutsch–Russell (MK–R) pathway. In this hybrid pathway, intermediates proceed down the Bloch pathway until demethylation of the sterol nucleus is complete, and then they undergo reduction of the double bond at C24 to enter the K–R pathway.

The step at which sterol synthesis crosses over from the Bloch to the K–R pathway could not be pinpointed in this experiment since some intermediates (shown in light gray in *Figure 1*) could not be measured due to either isobaric interference with cholesterol or because the levels were below the detection limits of the assay (see 'Material and methods'). Our data suggest that in SV-589 cells, sterols are demethylated via the Bloch pathway and then undergo Δ24 reduction upstream of desmosterol. Therefore, in cultured fibroblasts the cross-over to the MK–R pathway occurs at either zymosterol, dehydrolathosterol, or dehydrodesmosterol.

These experiments confirmed the conclusion of Kandutsch and Russell that significant differences exist between cells of different types in the pathways utilized for cholesterol synthesis (*Kandutsch and Russell, 1960a*, *1960b*); however, they did not provide evidence that the K–R pathway as originally conceived (*Figure 1*, red arrows) is utilized, at least in these two cell lines.

## Pathway utilization and cholesterol biosynthesis

To interrogate the factors that determine the relative utilization of the Bloch and MK–R pathway, we examined two cell culture systems in which rates of cholesterol biosynthesis were altered. First, we compared sterol flux in WT Chinese Hamster Ovary (CHO-7 cells) grown in cholesterol-containing medium (10% FCS) or in cholesterol-depleted serum (NCLPPS) to upregulate cholesterol synthesis. Second, cholesterol synthesis was examined in a mutant line of CHO-7 cells (SRD13A cells) that lack SREBP cleavage activating protein (SCAP), a protein required to activate the transcription of cholesterol biosynthesis genes (*Rawson et al., 1999*). SRD13A cells have no active SREBPs and thus exhibit low rates of cholesterol synthesis (*Rawson et al., 1999*). In WT CHO-7 cells, the Bloch pathway accounted for ~90% of cholesterol synthesis (*Figure 3A*). Deletion of SCAP reduced lanosterol biosynthesis by 80%, with similar reductions in the biosynthesis of other unsaturated side chain Bloch intermediates

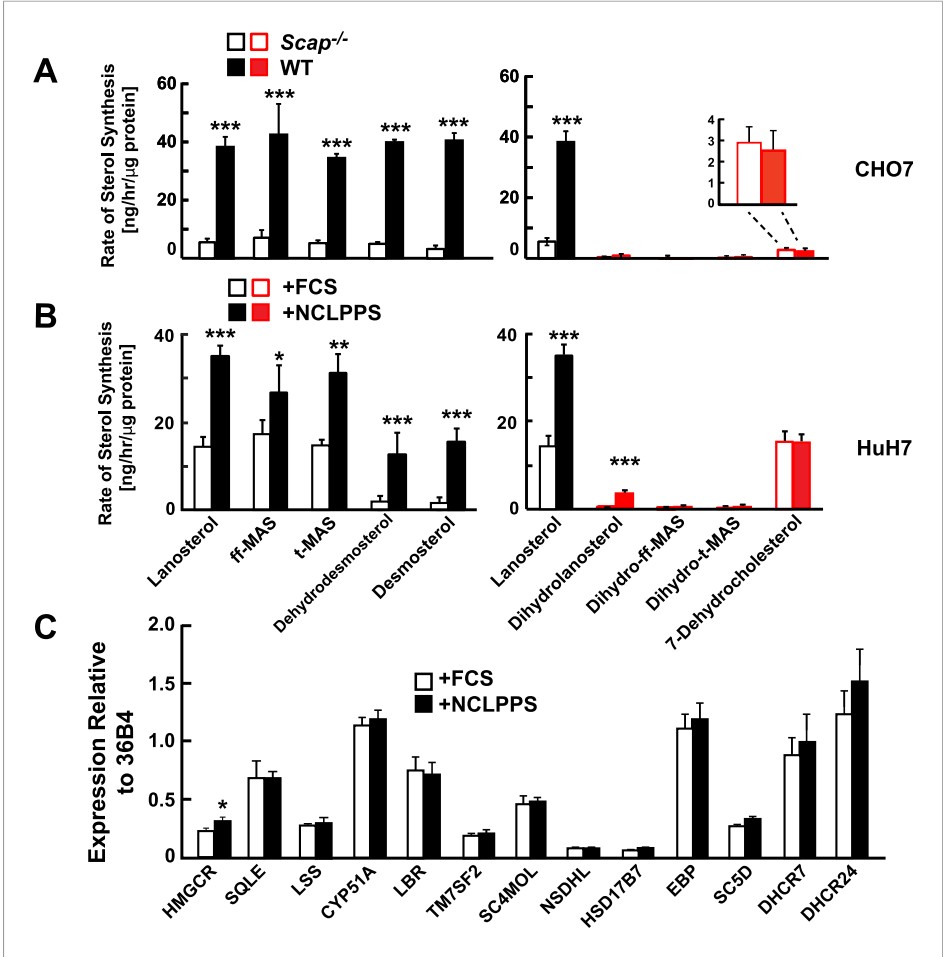

**Figure 3**. Modification of cholesterol biosynthesis rate. (**A**) Biosynthetic rate of intermediary sterols of the Bloch (left; black) and K–R (right; red) pathways of CHO-7 (wt, closed bars) and SRD13A cells (*SCAP$^{-/-}$*, open bars). Cells were grown to ~60% confluence in 10% NCLPPS before measuring sterol biosynthesis rates using $D_2O$. The inset of the right panel rescales the dehydrocholesterol values shown below. (**B**) Sterol biosynthesis rate in HuH7 cells grown in FCS (open bars) or the cholesterol depleted medium NCLPPS (solid bars). Cells were grown in their respective medium to ~60% confluence before $D_2O$ labeled to measure biosynthetic rates. In panels **A** and **B**, means and standard deviations are reported based on four independent replicates of each cell line or condition. (**C**) Expression of genes encoding enzymes catalyzing the conversion of squalene to cholesterol (shown in the Bloch sequence from left to right) in HuH7 cells grown in either FCS or NCLPPS. Expression is normalized to 36B4. Means and standard deviations are based on six replicates from two independent experiments. *p < 0.05, **p < 0.01, ***p < 0.001.

(*Figure 3A*). Despite the massive reduction in net sterol synthesis in the SRD13A cells, the biosynthetic rate of 7-dehydrocholesterol was similar to that observed in the parental cell line (*Figure 3A*, inset). This finding suggests the flux through the MK–R pathway is constitutive in these cells.

In cultured human hepatoma cells (HuH-7 cells) grown in cholesterol-replete serum (FCS) (*Figure 3B*), the pattern of sterol turnover resembled that seen in SV-589 cells (*Figure 2B*); most of the sterol flux traversed the MK–R biosynthetic pathway (*Figure 3B*, open bars). When cells were grown for 16 hr in cholesterol-depleted serum (NCLPPS) to stimulate cholesterol biosynthesis, sterol turnover doubled. Essentially all of the increase occurred via the Bloch pathway: flux through the MK–R pathway did not change (*Figure 3B*, open bars). Therefore, cholesterol depletion shifted the relative utilization of the two pathways such that approximately equal amounts of lanosterol were metabolized via Bloch and MK–R intermediates. Cholesterol depletion also resulted in the incorporation of a small amount of deuterium into dihydrolanosterol, presumably reflecting conversion from lanosterol, but none of the other K–R intermediates prior to 7-dehydrocholesterol were labeled in these cells.

To determine if the increase in Bloch pathway utilization associated with upregulation of cholesterol synthesis was coordinated at the transcriptional level, we compared levels of mRNAs encoding cholesterol biosynthetic enzymes in the HuH7 cells grown in FCS or NCLPPS (*Figure 3C*). Despite a significant increase in Bloch pathway utilization in the cells that were grown in NCLPPS, no corresponding changes were observed in expression of the post-squalene cholesterol biosynthetic enzymes, including DHCR24. Thus, the increase in flux through the Bloch pathway cannot be attributed simply to changes in the expression of SREBP-2 and the coordinated transcriptional upregulation of genes encoding enzymes in the pathway. Next, we examined the relationship between the level of expression of DHCR24, which desaturates the side-chain of the sterol, and the relative use of the Bloch and MK–R pathways.

## DHCR24 expression and pathway utilization

To determine if changing DHCR24 expression altered the relative use of the Bloch and MK–R pathways, we expressed recombinant mouse DHCR24 in human embryonal kidney cells (HEK-293 cells). In HEK-293 cells transfected with vector alone, 78% of cholesterol biosynthesis proceeded through the Bloch pathway (*Figure 4A*, open bars). Overexpression of DHCR24 resulted in lanosterol being quantitatively converted into the Bloch pathway intermediate zymosterol, but did not increase incorporation of label into the early methyl sterols of the classic K–R pathway. Only about one-third of the labeled zymosterol was converted to dehydrodesmosterol and desmosterol (*Figure 4A*, solid bars); however, flux through 7-dehydrocholesterol was markedly increased in these cells, indicating cross-over of post-zymosterol intermediates to the MK–R pathway. Taken together, these data indicate that overexpression of DHCR24 can promote flux through the MK–R pathway, thus diverting flux through the terminal half of the Bloch pathway. No evidence was seen of utilization of the classical K–R pathway under these conditions.

## Defining the point of crossover between the Bloch and MK–R pathways

As noted above, the flux of some sterol intermediates in the pathway could not be traced using deuterated water, which limited our ability to determine the point of cross-over from the Bloch to the MK–R pathway (*Figure 1*). To circumvent this problem, we incubated HEK-293 cells with synthetic isotopomers of lanosterol and zymosterol ($d_6$-lanosterol or $d_5$-zymosterol) that have isotopic distributions not found in natural sterols. By monitoring the conversion of these synthetic isotopomers into other sterols we were able to resolve all of the stable biosynthetic intermediate sterols in their labeled forms. In this experiment we were also able to examine cholesterol biosynthesis from desmosterol and from 7-dehydrocholesterol (*Figure 4B*, blue bars).

The design of these experiments differed from those shown previously. Here the labeled sterol was added to the medium and the distribution of label among the sterols of the two pathways was determined at a single time point (5 hr). In cells expressing vector alone, the $d_6$ label appeared in unsaturated side chain sterols of the Bloch pathway, with little to no label appearing in the MK–R intermediates (*Figure 4B*, open bars in upper panel). In cells expressing DHCR24, the $d_6$ label was detected in lanosterol, ff-MAS, t-MAS, and zymosterol, but not in the downstream Bloch intermediates (*Figure 4B*, closed bars in upper panel). Instead, the label appeared in the demethylated saturated side chain sterols along the MK–R pathway (zymostenol, lathosterol, and 7-dehydrocholesterol). These data suggest that zymosterol is the crossover point between the Bloch and MK–R pathways in these cells under these conditions, and that the methyl sterols upstream of zymosterol are poor substrates for (or do not have access to) DHCR24. Under both conditions, the majority of the labeled sterol remained as $d_6$-lanosterol. Overexpression of DHCR24 did not increase the formation of $d_6$-dihydrolanosterol or of $d_6$-cholesterol (*Figure 4B*, upper panel), suggesting the DHCR24 is not rate-limiting for the conversion of lanosterol to cholesterol.

To confirm the crossover point between the Bloch and K–R pathway, HEK-293 cells were incubated with $d_5$-zymosterol. In cells expressing the empty vector alone, $d_5$ label appeared in zymosterol, dehydrolathosterol, dehydrodesmosterol, and desmosterol (*Figure 4B*, open bars in lower panel). A small amount of label was measured in lathosterol and 7-dehydrocholesterol, but not in other K–R intermediates. In cells transfected with DHCR24, only a small fraction of the label was detected in intermediates of either pathway, presumably because they are rapidly converted to cholesterol. We found no label incorporated into dehydolathosterol or desmosterol. This result may reflect

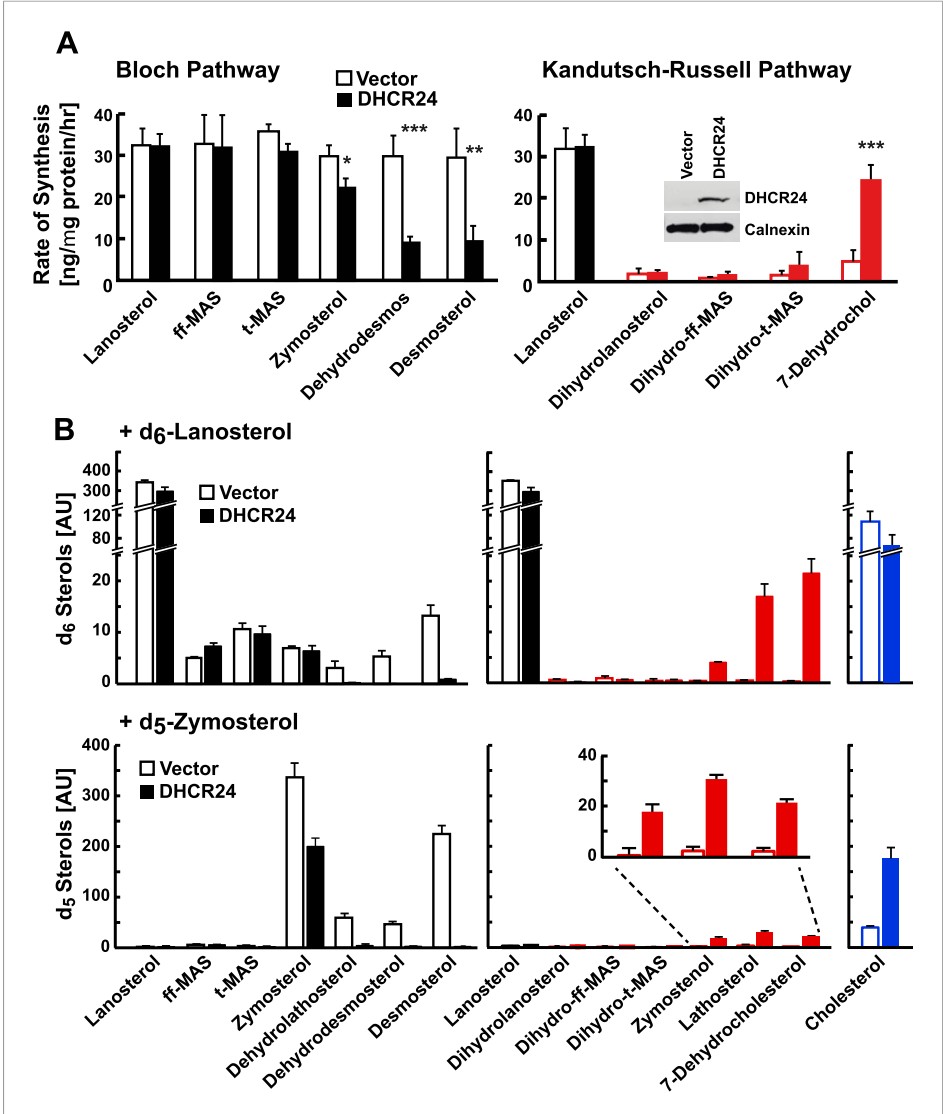

**Figure 4**. DHCR24 expression and sterol biosynthesis. (**A**) Sterol turnover in cells over-expressing DHCR24. HEK-293 cells were cultured in DMEM +10% FCS for 2 days. On day 3 the cells were transfected with empty vector (open bars) or a plasmid encoding DHCR24 (closed bars). After 40 hr, the medium was supplemented with 5% $D_2O$ and the incorporation of label into sterols was measured using LC-MS/MS. The rates of synthesis for intermediates in the Bloch (left) and K–R (right) pathways were determined using IA and fitted to a first order kinetic model. DHCR24 was measured by immunoblot analysis at the 0 hr time point (blot in right panel). Means and standard deviations shown are based on three independent experiments. (**B**) HEK-293 cells were transfected with DHCR24 (open) or an empty vector (closed). After 40 hr, 5 μg/ml of $d_6$-lanosterol (top) or $d_5$-zymosterol (bottom) conjugated to MCD was added to the medium. After 5 hr, cells were harvested and labeled sterols were measured by LC-MS/MS. Levels of labeled cholesterol are shown on the far right (blue). The inset of the bottom right panel highlights level of $d_5$-labeled zymostenol, lathosterol and 7-dehydrocholesterol. The signal intensities of the labeled sterols were normalized to total protein and to an internal standard. Means and standard deviations are based on 3 replicates. Similar results were observed in an independent experiment. *$p < 0.05$, **$p < 0.01$, ***$p < 0.001$.

quantitative conversion of zymosterol to zymostenol, or rapid conversion of desmosterol to cholesterol. Label was detected in the downstream MK–R pathway intermediates zymostenol, lathosterol, and 7-dehydrocholesterol (*Figure 4B*, closed bars in lower panel). These findings are consistent with those using labeled lanosterol and support the hypothesis that zymosterol is the first substrate for DHCR24 in the MK–R pathway used by these cells. The amount of $d_5$-cholesterol derived from $d_5$-zymosterol was 4 times higher when HEK-293 cells were transfected with DHCR24

relative to an empty vector (*Figure 4B*, right), suggesting that DHCR24 is rate-limiting for the conversion of zymosterol to cholesterol in these cells. No signal corresponding to $D_5$- or $D_6$- labeled sterols was seen in cells treated with vehicle alone.

## Cholesterol biosynthesis in the mouse

To examine the pathways of post-squalene cholesterol biosynthesis in vivo, we assessed rates of sterol synthesis in different mouse tissues. Mice were labeled to approximately 5% $D_2O$ by administering an initial bolus of $D_2O$ (500 μl) via intraperitoneal injection and by enriching their drinking water to 6% (vol/vol) $D_2O$. Rates of sterol synthesis were calculated as described in the 'Materials and methods' and *Figure 2—figure supplement 1*. The k-values and sterol concentrations are provided in *Supplementary file 1*.

In testes, deuterium label was only detected in Bloch pathway intermediates (*Figure 5*). The rates of synthesis of the methyl sterols (lanosterol, ff-MAS and t-MAS) in the Bloch pathway were ~3 times higher than those of the demethylated intermediates (zymosterol, dehydrolathosterol, dehydrodesmosterol and desmosterol) in the pathway. Thus, a large fraction of the t-MAS synthesized was diverted from the cholesterol biosynthetic pathway to produce other, as yet unidentified sterols (*Byskov et al., 1995*). The drop in flux along the Bloch pathway between t-MAS and zymosterol was not observed in any other tissue.

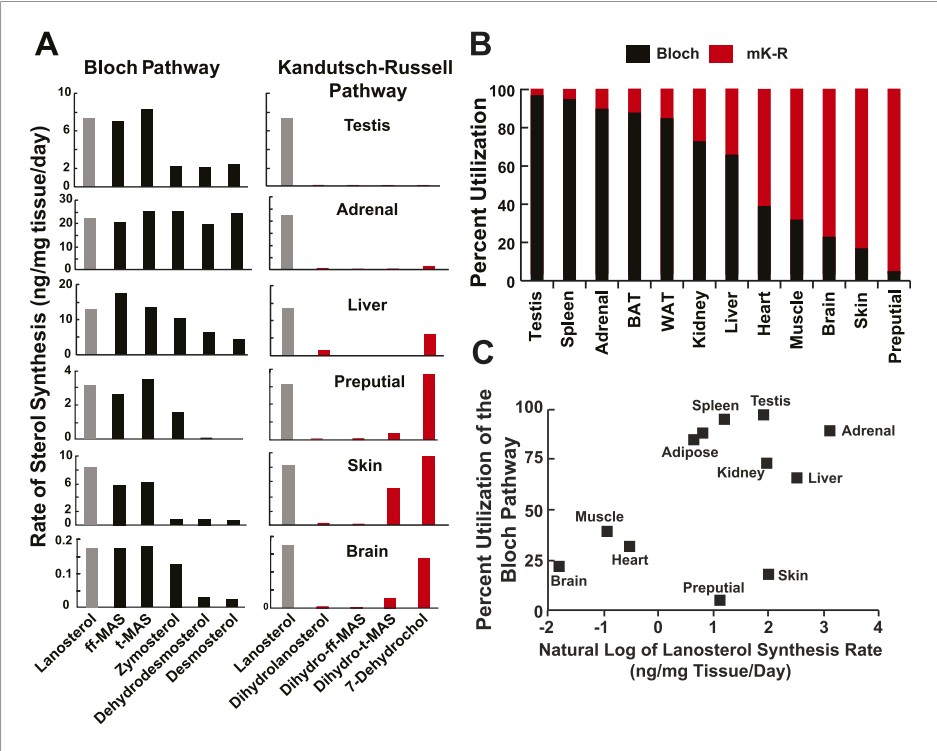

**Figure 5**. Sterol biosynthesis in mice. (**A**) Sterol biosynthesis in selected mouse tissues. Mice were enriched to ~5% $D_2O$ by intraperitoneal injection of 500 μl and supplementing the drinking water to 6% $D_2O$. Tissues were collected at 0, 1, 2, 3, 4, 6, 8, 12, 18, 24, 48, 72, 120, and 168 hr after injection (3 animals per time point), and analyzed by LC-MS/MS as described in the 'Materials and methods'. Rates of synthesis of the Bloch (left) and K–R (right) intermediates were calculated using IA and first-order kinetics. The experiment was repeated in an independent set of animals with similar results. (**B**) Percent utilization of the Bloch (black) and K–R (red) pathways for post-squalene cholesterol biosynthesis. The percentage of cholesterol derived from the Bloch pathway was determined by dividing the desmosterol synthesis rate by the sum of the desmosterol and 7-dehydrocholesterol synthesis rates. Similar results were obtained when the desmosterol synthesis rate was divided by the lanosterol synthesis rate, except in the testes (not shown). (**C**) The fractional utilization of the Bloch pathway using the data shown in Panel **B** compared to the natural log of lanosterol synthesis rate measured in each tissue.

In adrenal glands, cholesterol was also synthesized exclusively via the Bloch pathway (*Figure 5*) with little or no deuterium being incorporated into any of the saturated side chain intermediates of the K–R pathway. The rates of lanosterol and desmosterol synthesis were similar. Thus, lanosterol was not converted to any products other than desmosterol, and presumably cholesterol, at an appreciable rate.

In contrast to adrenal glands, the MK–R pathway predominated in skin. The synthesis of desmosterol was much less than either lanosterol or 7-dehydrocholesterol (*Figure 5*). There was measureable synthesis of dihydro-t-MAS, indicating that saturation of the side chain double bond can occur before total demethylation of t-MAS.

In liver, the rates of synthesis of desmosterol and 7-dehydrocholesterol were similar. Thus, approximately half of the sterol flux followed the Bloch pathway, whereas the other half used the MK–R pathway. In contrast to all other tissues examined, dihydrolanosterol was synthesized in liver at ~5–10% the rate of lanosterol synthesis (2 ng/µg tissue/day). Essentially no synthesis of dihydro-ff-MAS was detected. This result suggests that a fraction of hepatic lanosterol was converted to dihydrolanosterol but did not proceed down the K–R pathway. Consistent with this finding, cultured human hepatocytes (HuH7 cells) also synthesized dihydolanosterol that was not further metabolized into dihydro-ff-MAS (*Figure 3B*). Thus ~5–10% of the sterol synthesized in liver or in cultured hepatocytes is converted to dihydrolanosterol and exits the cholesterol biosynthetic pathway.

We performed similar in vivo analyses on preputial gland and brain (*Figure 5*). The K–R pathway was first described in preputial gland, a modified sebaceous gland that is present in rodents but not in humans (*Kandutsch and Russell, 1960a*, *1960b*). We found that the turnover of 7-dehydrocholesterol in preputial glands was similar to that of lanosterol, which is consistent with the findings of Kandutsch and Russell (2). Nevertheless, our data are not consistent with the K–R pathway as it was originally described. The flux from lanosterol proceeded via the Bloch pathway intermediates ff-MAS and t-MAS without any detectable incorporation into the corresponding saturated side chain sterols (dihydrolanosterol and dihydro-ff-MAS). Therefore, even in preputial glands, lanosterol demethylation commences before side-chain saturation.

A similar pattern of sterol flux through the MK–R pathway was measured in the skin and brain. The skin makes Vitamin D, but since we did not measure incorporation of label into cholesterol we do not know how much of the 7-dehydrocholesterol that is made in the dermis is converted to Vitamin D vs. cholesterol. In the brain, the absolute rate of cholesterol synthesis was low, corresponding to less than 2% of that observed in liver (*Figure 5*, bottom panel; note differences in scale). Like the skin and preputial gland, the MK–R pathway for cholesterol synthesis predominated in the central nervous system, even though the brain was reported previously to express very low levels of DHCR24 (*Nes, 2011*).

Thus, none of the tissues examined in this study utilized the classic K–R pathway. Instead, sterols were demethylated, at least partially, before the side chain was saturated.

To compare the relative pathway utilization across tissues, the fractional utilization of the Bloch pathway was estimated for each tissue by dividing the rate of synthesis of desmosterol by the sum of the rates of synthesis of desmosterol and 7-dehydrocholesterol. The values obtained ranged from 0.97 (testes) to 0.08 (preputial gland) (*Figure 5B*). In general, tissues with a higher lanosterol synthesis rate had a higher fractional utilization of the Bloch pathway (*Figure 5C*), with the exception of the skin and preputial gland, which had high sterol synthesis rates via the MK–R pathway.

## Imperfect correlation between DHCR24 expression and use of MK–R pathway

It has been suggested that relative usage of the Bloch and K–R pathway is determined by level of expression of DHCR24 (*Nes, 2011*). To determine if the tissue-specific differences we observed in usage of the MK–R pathway were caused by differences in DHCR24 expression, we measured levels of DHCR24 mRNA and protein in the different tissues (*Figure 6*). The tissue with the highest level of both DHCR24 mRNA and protein was the preputial gland, which also has highest fractional utilization of the MK–R pathway (*Figure 5*). Surprisingly, the tissue with the second highest level of DHCR24 mRNA and protein was the liver, which predominantly utilizes the Bloch pathway (*Figure 5A*). Skin, which has the second highest fractional usage of the MK–R pathway for cholesterol biosynthesis, had mRNA and protein levels of DHCR24 that were lower than those found in the liver. In the brain the level of DHCR24 mRNA was higher than that detected in the skin and yet no protein was detected in this tissue. Thus, the level of DHCR24 mRNA in tissues did not always correlate with the fractional utilization of the MK–R pathway.

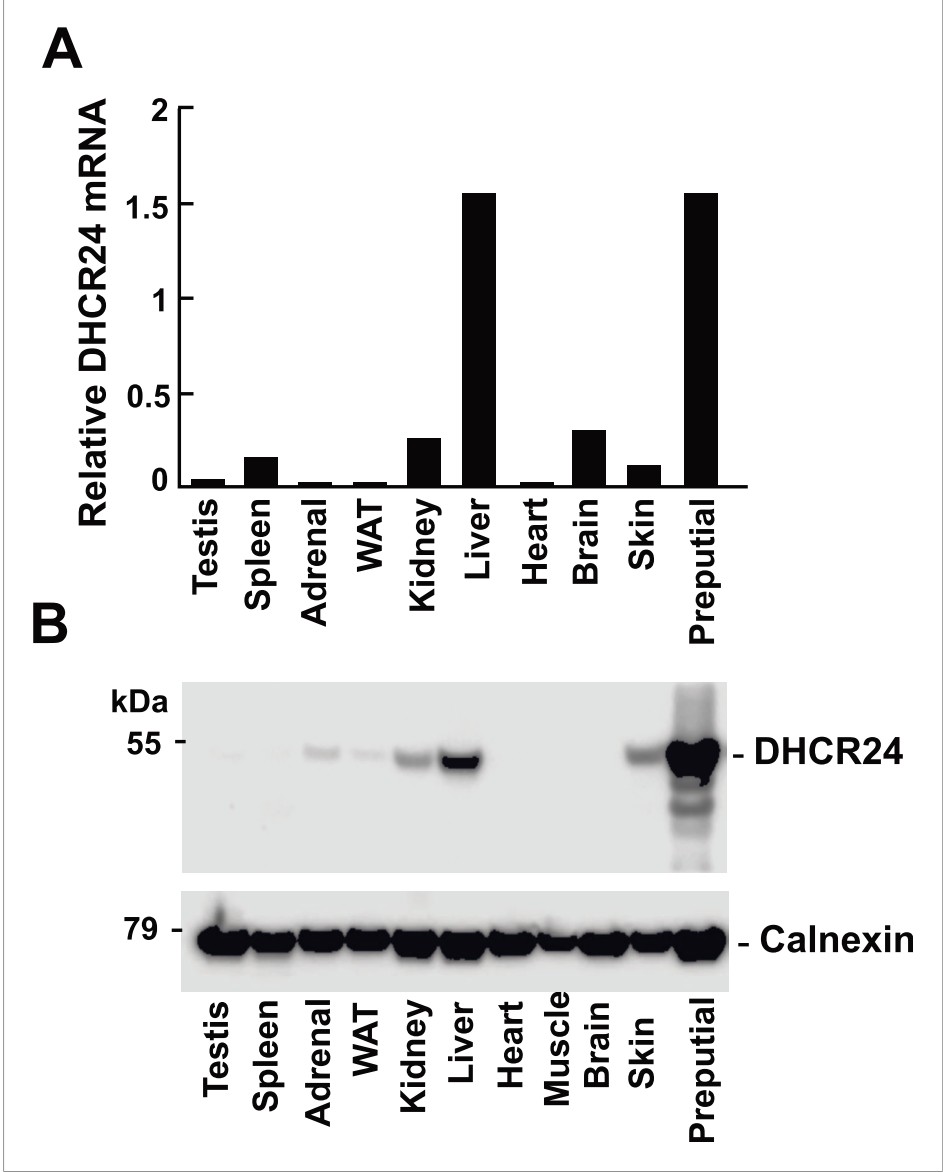

**Figure 6**. Levels of DHCR24 mRNA and protein in mouse tissues. (**A**) The levels of DHCR24 mRNA relative to 36B4 in mouse tissues are arranged from highest to lowest fractional utilization of the Bloch pathway (left to right). (**B**) Immunoblot analysis of DHCR24. Tissues were collected from a single male mouse and a total of 5 µg of protein from each tissue was analyzed by immunoblotting using a polyclonal rabbit anti-mouse antibody as described in the 'Material and methods'. The experiment was performed 3 times with similar results.

In general, DHCR24 was expressed at very low levels in tissues that predominantly use the Bloch pathway (testes, spleen, adrenal and adipose tissue) and in those with low cholesterol biosynthetic rates (heart and muscle) (*Figure 5C*).

## Turnover of sterols in plasma reflects the turnover of sterols in liver

In mammals, liver is the major source of plasma cholesterol (*Dietschy and Turley, 2002*). To assess the feasibility of inferring rates of hepatic sterol biosynthesis from plasma samples, we simultaneously measured the turnover of sterols in the plasma and liver of mice (*Figure 7*). We found that the fractions of sterols with incorporated label in plasma and liver were remarkably similar for lanosterol, desmosterol, and 7-dehydrocholesterol (*Figure 4C*). The only sterol for which the labeling pattern was substantially different between the two compartments was dihydrolanosterol, which readily

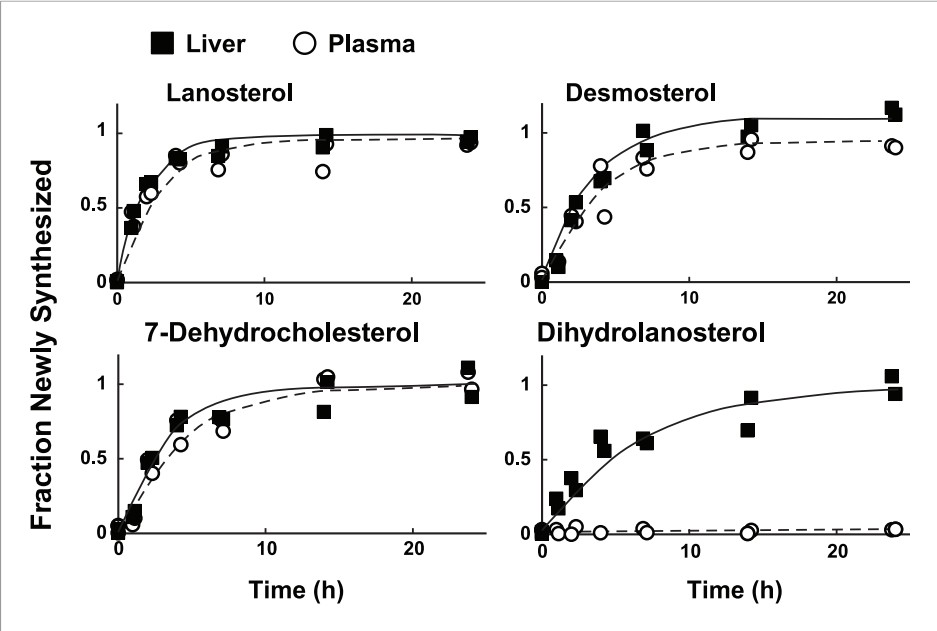

**Figure 7**. Sterol turnover in liver and plasma of mice. The fractional turnover of lanosterol, desmosterol, 7-dehydrocholesterol, and dihydrolanosterol was determined based on the incorporation of deuterium into sterols measured in liver (closed boxes) and plasma (open circles) from the same mice.

incorporated label in liver but remained unlabeled in plasma. We were unable to determine whether dihydrolanosterol was being converted to another sterol, was quantitatively excreted into bile and thus did not enter the plasma compartment, or was obscured in the MS assay by an analyte that is present in plasma, but not liver.

## Discussion

The Bloch and K–R pathways were described more than 50 years ago and are widely accepted as the two major pathways for cholesterol synthesis (*Kandutsch and Russell, 1960b*; *Bloch, 1965*). The present study represents the first systematic analysis of flux through these pathways in cell culture and in vivo. The major finding of the study is that the architecture of the cholesterol biosynthetic pathway shows striking differences among tissues. The testes and adrenal gland utilize the canonical Bloch pathway (*Bloch, 1965*) almost exclusively (*Figure 8*, black arrows). None of the tissues examined in this study had a pattern of sterol turnover consistent with the reaction sequence proposed by *Kandutsch and Russell (1960b)* (*Figure 1*, red arrows). A hybrid pathway that we have named the MK–R pathway exists in skin, preputial glands and brain, (*Figure 8*, red arrows). In these tissues, sterols undergo demethylation prior to side chain saturation (*Figure 5*). Whereas we were not able to localize the specific DHCR24 substrate in the $D_2O$ labeling studies, experiments using deuterium-labeled lanosterol and zymosterol in HEK-293 cells confirmed that zymosterol is the first sterol to undergo appreciable side-chain reduction, at least in these cells. Remarkably, immortalized cells in culture retained the specific pathways used by their tissues of origin. Regulation of cholesterol biosynthesis was achieved almost exclusively by changes in flux through the Bloch pathway. Our study reveals a strikingly varied pattern of sterol synthesis from lanosterol that allows for the generation of multiple, tissue-specific end-products in addition to cholesterol.

The MK–R pathway is consistent with a reaction sequence proposed by *Bae and Paik (1997)*, who reported that zymosterol was preferred over lanosterol as a substrate for C-24-reduction in rat liver microsomes. Those authors hypothesized that the physiological pathway for conversion of lanosterol to cholesterol did not correspond to either the Bloch or the KR pathways, but rather to a reaction sequence in which C24-reduction occurred after demethylation but before the final Δ5-dehydrogenation and Δ7-reduction of the sterol nucleus. This proposal was not assessed in

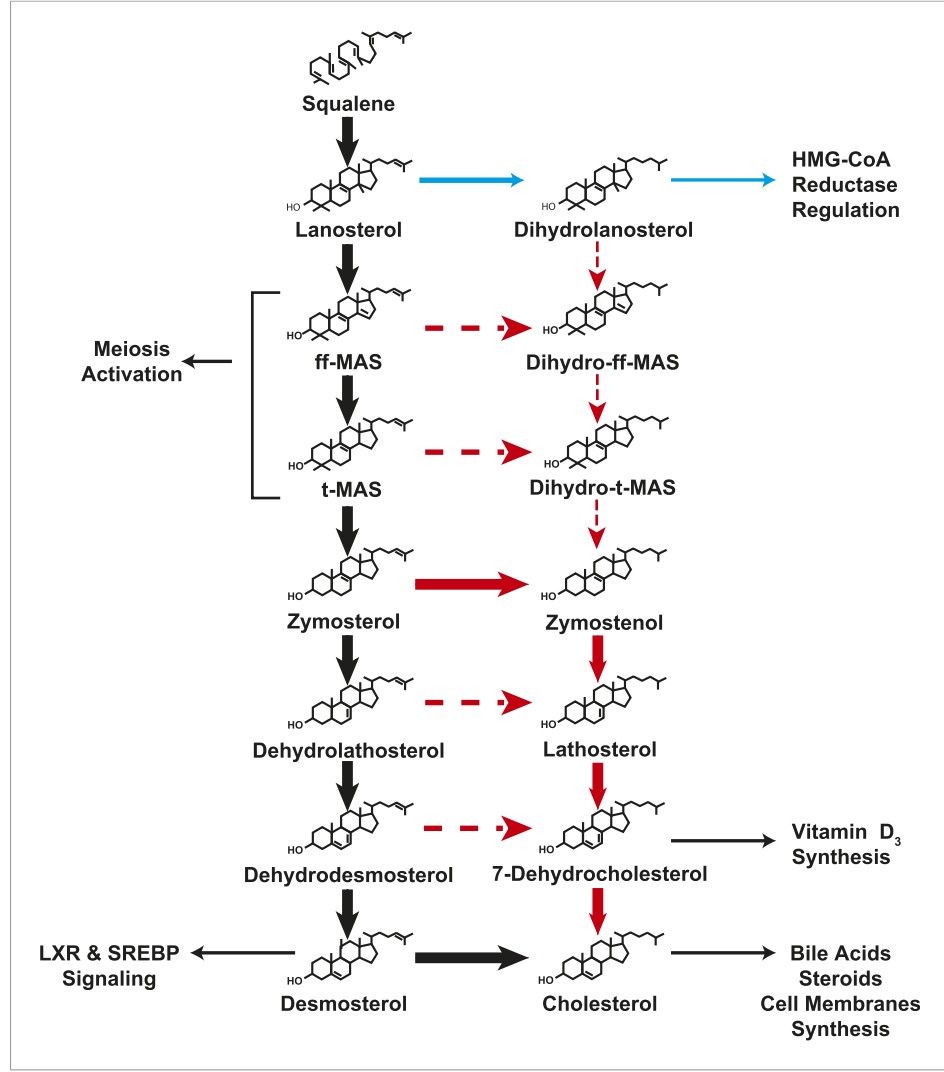

**Figure 8**. A modified Kandutsch–Russell (MK–R) model of post squalene cholesterol biosynthesis. In this model, cholesterol biosynthesis proceeds from lanosterol to t-MAS. Black arrows denote the Bloch pathway. Red arrows denote the MK–R pathway. 24,25-double bond desaturation can occur at any step between lanosterol and desmosterol in this pathway, but in most tissues desaturation does not occur until after demethylation is complete (after T-MAS). The pathway shown with blue arrows was only detected in the liver and did not contribute to cholesterol biosynthesis. Additional pathways involving sterol intermediates that do not result in the biosynthesis of cholesterol are also shown.

vivo and did not gain wide acceptance. The present data provide in vivo evidence for 24-reduction of zymosterol, but the predominance of the Bloch pathway in most tissues with high rates of cholesterol synthesis indicates that desmosterol is a major physiological substrate for the enzyme.

Reduced cellular cholesterol content selectively upregulated the Bloch pathway in cultured CHO7 and HuH7 cells (*Figure 3*). Genetic ablation of SCAP, which is required for SREBP activation (*Rawson et al., 1999*), dramatically reduced flux through the Bloch pathway, but had virtually no effect on the MK–R pathway. Similarly, depletion of cellular cholesterol by incubation in NCLPPS resulted in a major increase in flux through the Bloch pathway with no detectable change in flux through the MK–R pathway. These findings suggest that the MK–R pathway is constitutively active and that the Bloch pathway is used preferentially for regulated cholesterol biosynthesis in response to fluctuations in cholesterol availability and demand. Upregulation of the Bloch pathway also increases the synthesis of regulatory sterols that limit cholesterol accumulation in the cells: desmosterol activates LXR, which

promotes cholesterol efflux from cells, and acts as a feedback inhibitor of cholesterol synthesis by inactivating SREBP (*Yang et al., 2006*; *Spann et al., 2012*).

The testes and adrenal glands synthesized cholesterol predominantly via the Bloch pathway, but the fate of sterols moving through the pathway in the two organs differed markedly. In the adrenal glands, lanosterol was quantitatively converted to desmosterol, and then presumably to cholesterol, which provides the substrate for adrenal steroidogenesis. In contrast, only one-third of the lanosterol synthesized in the testes was converted to desmosterol, and virtually none entered the MK–R pathway. In this tissue, lanosterol was quantitatively converted to ff-MAS and then to t-MAS, but more than two-thirds of the t-MAS was diverted from the pathway before zymosterol (*Figure 5*). The metabolic fate of t-MAS in the testes is not known. t-MAS and its immediate precursor ff-MAS have been implicated as meiosis-activating sterols in the formation of male and female germ cells (*Byskov et al., 1995*); however, the role of these precursor sterols remains controversial. Germ-cell specific ablation of *Cyp51a1,* the enzyme that converts lanosterol to ff-MAS (*Figure 1*), markedly reduced testicular t-MAS concentration but had no effect on reproductive function in male mice (*Keber et al., 2013*). The reduction in flux from t-MAS to zymosterol that we observed in mouse testes (*Figure 5*) may be due to t-MAS being converted to another sterol (or steroid hormone), or to the secretion of t-MAS from the testes.

The liver was the only tissue in which we observed appreciable dihydrolanosterol synthesis (*Figure 5*). Approximately 5–10% of the lanosterol synthesized in the liver was converted to dihydrolanosterol, but not to downstream intermediates in the K–R pathway. In contrast to other sterol intermediates synthesized in the liver, labeled dihydrolanosterol did not appear in the plasma (*Figure 7*). It is possible that dihydrolanosterol is rapidly excreted from the liver via the bile. Although we did not include biliary sterols in our flux studies, we have found that the concentration of dihydrolanosterol in bile is not enriched relative to the liver (data not shown). This finding is not compatible with selective biliary excretion of dihydrolanosterol. Taken together, our data indicate that at least some of the dihydrolanosterol that is formed in the liver is converted to another sterol, the identity of which remains unknown.

The function of dihydrolanosterol in the liver is also not known. Previous studies have demonstrated that dihydrolanosterol can promote degradation of HMG-CoA reductase (*Lange et al., 2008*). This effect is specific to dihydrolanosterol: neither cholesterol nor the other biosynthetic intermediates tested, including lanosterol and lathosterol, showed this activity (*Song et al., 2005*; *Lange et al., 2008*). Therefore, one function of the dihydrolanosterol (or a metabolic derivative thereof) synthesized in the liver may be to provide rapid feedback inhibition of cholesterol biosynthesis through post-transcriptional regulation of HMG-CoA reductase. If this model is correct, then our data suggest that this regulation is likely only significant in liver.

Three tissues predominantly used the MK–R pathway: brain, skin and preputial gland. Our finding that skin and preputial gland produced high levels of saturated side chain sterols is compatible with the findings of *Kandutsch and Russell (1960b)*. The utilization of the MK–R pathway in the skin ensures a constant supply of 7-dehydrocholesterol for vitamin D synthesis (*DeLuca, 2008*). Why the MK–R is the predominant pathway in preputial gland is less clear. The major function of this gland is thought to be the synthesis of pheromones. The chemical nature of these pheromones has not been fully defined but squalene and cholesterol were the most abundant lipids identified in a GC/MS analysis of preputial glands from male rats (*Zhang et al., 2008*). Thus, cholesterol, or one or more of its biosynthetic precursors may be used for pheromone synthesis. In the present study, a loss of sterol between lanosterol and 7-dehydrocholesterol was not observed, but we cannot exclude the possibility that some 7-dehydrocholesterol is diverted from cholesterol synthesis to pheromone synthesis.

In contrast to skin and preputial gland, the brain has very low DHCR24 activity and was predicted to predominantly use the Bloch pathway (*Nes, 2011*). Although the brain is among the most cholesterol-rich organs, turnover, and hence synthesis of brain cholesterol is low (*Figure 5*) (*Lund et al., 2003*). Our data indicate other tissues with low rates of cholesterol synthesis, such as heart and skeletal muscle also predominantly used the MK–R pathway (*Figure 5*). These tissues are largely post-mitotic and do not synthesize steroid hormones or vitamin D. If the MK–R pathway is constitutive in vivo, as it appears to be in cultured cells, then predominant use of this pathway may ensure a constant rate of cholesterol synthesis in tissues that experience little variation in their cholesterol requirements.

The relative utilization of the Bloch and MK–R pathways in cultured cells was strikingly similar to that observed in vivo in the tissue of origin. These findings suggest that the factors governing pathway selection are epigenetically fixed. Relative pathway utilization is presumably a function of the relative activities of the two enzymes at the branch point, which is tissue dependent. The relative function of the two enzyme could be determined by myriad factors including those that act directly on enzyme activity (e.g., transcription, post-translational modification), the expression of auxiliary proteins (co-factors, inhibitors, proteins that present or sequester substrate) or factors that affect the relative proximity of the enzymes to their substrates.

Since DHCR24 activity is a major determinant of relative pathway utilization, cell-type specific control of pathway utilization may involve epigenetic control of DHCR24 expression. The pattern of DHCR24 expression in tissues was an imperfect predictor of relative pathway utilization. For example, the brain used the MK–R pathway despite low DHCR24 expression and activity (*Tint et al., 2006*) while liver and kidney predominantly used the Bloch pathway, despite relatively high expression of DHCR24 (*Figure 6*). DHCR24 is phosphorylated at multiple sites and may be regulated at the post-translational level (*Luu et al., 2014*). The activity and substrate specificity of the enzyme may also be influenced by its intracellular location. Brown and his colleagues reported that DHCR7 and DHCR24 co-immunoprecipitate in CHO-7 cells (*Luu et al., 2015*). Those authors proposed that the formation of a complex between the two proteins favors the efficient conversion of 7-dehydrodesmosterol to desmosterol, and then to cholesterol. This hypothesis is consistent with our finding that the Bloch pathway is predominantly used in steroidogenic cells. Further studies are required to elucidate the mechanistic basis for the wide differences in DHCR24 expression among tissues, and to identify other factors that govern the route of flux through the cholesterol biosynthetic pathways.

The patterns of deuterium enrichment of biosynthetic precursor sterols in plasma were similar to those seen in the liver, but distinct from those observed in extrahepatic tissues. This finding indicates that the cholesterol pool in extrahepatic tissues is relatively isolated from the cholesterol in circulating lipoproteins. This hypothesis is supported by data from Dietschy and colleagues, who reported that 80% of cholesterol biosynthesis in mice takes place in extra-hepatic tissues, while 80% of plasma low density lipoprotein cholesterol is taken up by the liver (*Osono et al., 1995*). Thus, most extrahepatic tissues obtain cholesterol primarily from de novo synthesis, with little contribution from circulating lipoproteins. This conclusion suggests that the effects of dietary and pharmacological interventions on cholesterol biosynthesis in the liver can be inferred from measurements of isotopic enrichment in the plasma.

If extrahepatic tissues contribute little to the plasma sterol pool as shown in this study, how are sterols synthesized in peripheral tissues transported to the liver (or gut) for excretion? The route by which cholesterol is transported from the peripheral tissues to the liver, a process termed reverse cholesterol transport (*Glomset, 1968*), remains to be fully defined (*Hellerstein and Turner, 2014*) One possibility is that peripheral tissues release cholesterol, but not the biosynthetic precursor sterols, into the circulation. In this case we would fail to capture flux of peripheral tissue cholesterol since we did not measure cholesterol isotopomers in these experiments. An alternative possibility is that cholesterol from the periphery is transported in the circulation by cells rather than lipoproteins or other plasma components. Careful quantification of cholesterol fluxes in vivo will be required to elucidate the relative contributions of cellular and plasma elements to the centripetal transport of peripheral tissue cholesterol.

The approach used in this study could potentially be confounded by the presence of two pools of an intermediate in the pathway that turn over at different rates. This problem would be particularly acute if a small pool of intermediate turns over rapidly while a second, larger pool sequestered from the biosynthesis pathway turned over slowly. Under these conditions the larger pool may significantly reduce the labeling of the total intermediate isolated from the cells resulting in falsely low estimates of flux. The observation that multiple independent intermediates showed essentially identical kinetics argues against this possibility, but we cannot formally exclude it.

Metabolic pathways have traditionally been defined in two stages. First, the sequence of reactions that comprise the pathway is determined biochemically. These determinations are almost invariably qualitative in nature. Second, flux through the pathway is quantified using radioisotopes to trace the turnover of a single metabolite, typically the end product of the pathway. By combining LC-MS/MS methods with stable isotope tracing, flux can be monitored through multiple intermediates in a biosynthetic pathway simultaneously. As illustrated in the present study, this approach can reveal

bifurcations in established pathways that imply novel biochemistry and physiology for molecules previously viewed simply as biosynthetic intermediates.

## Materials and methods

Deuterium labeled water ($D_2O$, 99.8% pure) was purchased from Sigma. Cholesterol, lanosterol, dihydrolanosterol, ff-MAS, t-MAS, zymosterol, desmosterol, 7-dehydrocholesterol, $d_5$-zymosterol, $d_6$-lanosterol, and $d_6$-sitosterol were obtained from Avanti Polar Lipids (Alabaster, AL). 25-hydroxycholesterol (25-OHC) was obtained from Steraloids, Inc. (Newport, RI). Methyl-cyclodextrin (MCD) was obtained from Cyclodextrin Technologies (High Springs, FL). $d_5$-zymosterol, $d_6$-lanosterol were bound to MCD in a 1:10 and 1:20, respectively, stoichiometric ratio for solubilization as previously described (*Brown et al., 2002*).

### Cell culture

Immortalized cells from human fibroblasts (SV-589 cells) (*Yamamoto et al., 1984*), human hepatoma (HuH-7 cells) (*Nakabayashi et al., 1984*), mouse adrenocortical tumor (Y1-BS1 cells) (*Watt and Schimmer, 1981*), human embryonic kidney (HEK-293 cells) (*Graham et al., 1977*), Chinese hamster ovaries (CHO-7 cells), and SREBP cleavage activation protein deficient CHO-7 (SRD13A; *SCAP$^{-/-}$*) cells were used in this study (*Rawson et al., 1999*). Cells were plated at a density of 500,000 cells per 60 mm dish and maintained in monolayer culture at 37°C in 5% $CO_2$. SV-589, HEK-293, CHO-7, SRD13A and HuH-7 cells were grown in Dulbecco's modified Eagle's medium (DMEM) supplemented with either 10% (vol/vol) fetal calf serum (FCS) or newborn calf lipoprotein poor serum (NCLPPS). Y1-BS1 cells were grown in a 1:1 mixture of Ham's F-12 medium and DMEM supplemented with 15% horse serum (HS). All media contained 100 units/ml penicillin and 100 μg/ml streptomycin sulfate. In experiments using NCLPPS and/or 25-OHC, the medium was changed 16 hr before beginning $D_2O$ labeling.

For measurements of sterol synthesis, cells were plated and grown to ~60% confluence either 2 days (SV-589, CHO-7, SRD13A, and HEK-293), 3 days (HuH7), or 5 days (Y1BS1). Then the medium was exchanged with medium containing 5% $D_2O$. After 0, 0.5, 1, 2, 4, 6, 8, 12, and 24 hr, cells (triplicate dishes at each time point) were washed and then harvested in 3 ml Dulbecco's phosphate buffered saline (PBS) with 0.5% Triton X-100. A 300 μl aliquot of each cell lysate was reserved for measurement of protein concentration by bicinchoninic acid assay (BCA) analysis (Pierce BCA Protein Assay Kit). The remaining cell lysate was mixed with 3 ml methanol (MeOH) containing 20 ng of $d_6$-sitosterol as an internal standard. The samples were sonicated for 5 min and stored at room temperature until they were prepared for LC-MS/MS analysis (see below).

### Mice

Male C57BL/6J mice were obtained from the Jackson Laboratory (Bar Harbor, ME), housed (4 per cage) in a controlled environment (12-hr light/12-hr dark daily cycle, 23 ± 1°C, 50–70% humidity) and fed *ad libitium* with standard chow (Harlan; Teklad, 2016) for 4 weeks prior to experimentation. 3 days prior to experimentation the mice were entrained to a synchronized feeding cycle (12 hr fasting, 12 hr refeeding) by removing food at the beginning of the light cycle and returning food at the onset of the dark cycle. All research protocols involving mice were reviewed and approved by the Institutional Animal Care and Use Committee at University of Texas Southwestern Medical Center.

Sterol synthesis rates in mice were determined by measuring deuterium incorporation from $D_2O$ over time. At 9 AM (3 hr into the light cycle) each mouse was injected intraperitoneally with 500 μl of $D_2O$ with 150 mM NaCl. After injection, the drinking water was supplemented with 6% $D_2O$. Mice (3 per time point) were sacrificed at 1, 2, 3, 4, 6, 8, 12, 18, 24, 48, 72, 120, and 168 hr after injection. Six mice that did not receive $D_2O$ were also sacrificed to measure the concentrations of sterols in their tissues. Immediately after sacrifice the adrenals, blood, brain, brown adipose tissue (BAT) from the nape of the neck, heart, kidneys, liver, preputial glands, skin, spleen, posterior hind limb muscle, testes, and epididymal adipose tissue (WAT) were removed. Plasma was isolated from blood obtained in an EDTA-coated tube after centrifugation. Fur was removed from the skin by applying Veet, thoroughly washing in PBS to remove residue, and then placed at −80°C. All other tissues were snap frozen in liquid $N_2$ and stored at −80°C.

## Preparation of samples for LC-MS/MS

Larger tissues were prepared for lipid extraction by weighing 50–100 mg pieces and immediately adding 5 ml of ice cold MeOH and 200 ng of $d_6$-sitosterol. Tissues were thoroughly homogenized using an IKA pole rotor and bath sonication for 10 min at RT. The sample was then vortexed and centrifuged at 3000×$g$ for 10 min. The supernatant was decanted and the pellet was resuspended in 5 ml MeOH and recentrifuged. The supernatant was pooled with the supernatant from the initial centrifugation. 1 ml of supernatant was removed and diluted with 2 ml of MeOH and 3 ml of PBS. Due to their small size, the adrenal and preputial glands were prepared by weighing the entire tissue, and then adding 3 ml MeOH and 20 ng of $d_6$-sitosterol. The tissues were then homogenized and 3 ml PBS was added. Plasma was prepared by adding 100 µl to 6 ml of 1:1 MeOH/PBS while sonicating and adding 20 ng $d_6$-sitosterol. From this point forward, cell culture, tissue, and plasma samples were handled identically.

Lipids were extracted from the samples by a modified Bligh-Dyer extraction (*Bligh and Dyer, 1959*). Samples were saponified by adding 300 µl 45% (wt/vol) KOH and incubation for 2 hr at 60°C. After saponification, the samples were allow to cool to RT before 3 ml of dichloromethane (DCM) was added, inducing a 2-phase separation. The samples were vortexed and centrifuged at RT. The bottom phase (containing primarily DCM) was removed. An additional aliquot of DCM (4 ml) was added to the top phase, vortexed, centrifuged, and the bottom phased was pooled with the previous bottom phase. The samples were evaporated under a light stream of $N_2$, resuspended in 300 µl of 9:1 MeOH/$H_2O$, and transferred to vials for LC-MS/MS analysis.

## LC-MS/MS analysis

Sterols were analyzed as described by *McDonald et al. (2012)*. Lipid extracts were injected into a Shimadzu LC20A HPLC (Kyoto, Japan) with an Agilent Poroshell 120 EC-C18 column (2.1 × 150 mm, 2.7 micron beads, Santa Clara, CA) and eluted using a solvent gradient that transitioned linearly from 93% MeOH/7% $H_2O$ to 100% MeOH in 7 min. The column was washed for 5 min in 100% MeOH and then returned to the initial solvent. Sterols were detected using an ABSciex (Framingham, MA) 4000 Qtrap MS/MS in positive mode with atmospheric pressure chemical ionization at a temperature of 350°C. The MS/MS detected mass to charge ratios ($m/z$) of 365–370, 393–400, 404, and 409–414, which spans the ion $m/z$ plus 3 mass units for each sterol, along with the internal standard (see *Supplementary file 2* for details). For experiments that involve pre-labeled sterols, the mass ranges were extended to measure the M+5 and M+6 isotopomers.

Standards were commercially available for all but four of the sterols in the cholesterol biosynthetic pathway. These standards were used for quantitation of sterol concentrations relative to $d_6$-sitosterol, which was added as an internal standard. The four sterols that were not commercially available—dihydro-ff-MAS, dihydro-t-MAS, dehydrolathosterol, and dehydrodesmosterol—were identified by their unique $m/z$ values and retention times. Based on their chemical structures, the $m/z$ of dihydro-ff-MAS, dihydro-t-MAS, dehydrolathosterol, and dehydrodesmosterol are predicted to be 397, 395, 367, and 365 Da, respectively. The retention times of these sterols was determined by analyzing sterol spectra of liver and feces from mice (which have high concentrations of cholesterol biosynthetic intermediates relative to cholesterol) consuming water supplemented with 10% $D_2O$ for 6 weeks. Sterol spectra in the $D_2O$ labeled mice were compared to those of unlabeled mice to identify peaks of the correct molecular mass that had an MS/MS fragmentation pattern characteristic of sterols and were endogenously synthesized (based on deuterium incorporation). Only a single chromatographic peak met these criteria for each sterol. This strategy was tested using two additional sterols, ff-MAS and zymostenol, which were correctly identified when compared to the authentic standard.

Three biosynthetic intermediates were not measured in this study. The concentrations of dehydrolathosterol were too low in all tissues and cell culture lines to reliably measure the isotopomer distribution. Lathosterol and zymostenol cannot be reliably measured in this system because they are isobaric with cholesterol, which saturates the signal at the $m/z$ value of 369 Da. Cholesterol also saturated the isotopomers greater than M+1 for sterols with $m/z$ = 367, including desmosterol, 7-dehydrocholesterol, and zymosterol.

## Data interpretation

Individual peaks for each sterol in *Supplementary file 2* were integrated with Analyst software (Framingham, MA). For each sterol, the fractional contribution of M+0 isotopomer (M0$_m$) was

calculated by dividing the intensity of the M+0 peak by the summed intensity of all measureable isotopomers of that sterol (M+0, M+1, M = 2…). The fraction of each sterol that was newly synthesized (g) at each time point was determined by linearly deconvoluting $M0_m$ based on the fractional contribution of M+0 of natural ($M0_n$; ~0.67) and fully labeled ($M0_t$; ~0.35) sterols, expressed as:

$$M0_m = gM0_t + (1 - g)M0_n.$$

Which was solved for g:

$$g = \frac{M0_m - M0_n}{M0_t - M0_n}.$$

This approach, which is called ISA or mass isotopomer distribution analysis (MIDA), was developed by *Kelleher and Masterson (1992)* and *Hellerstein and Neese (1992)* (see *Figure 2—figure supplement 1* for a schematic representation).

$M0_n$ was determined from the isotopic distributions of unlabeled samples (*Figure 2A*), which were in good agreement with theoretical distributions based on the natural abundances of $^{13}C$, deuterium, and $^{18}O$. The value of $M0_t$ is dependent on both the enrichment of the labeling pool (p; i.e., the fraction of water that is $D_2O$) and the number of incorporation sites for label per molecule (N; i.e., the potential sites for deuterium incorporation). $M0_t$ was initially determined based on the experimentally determined asymptotic spectrum of isotopomers at the last time point of labeling (*Figure 2A*) (i.e., the distribution of isotopomers when all sterols were synthesized in the presence of $D_2O$). The value was refined using MIDA as described by *Hellerstein and Neese (1992)*. For cell culture experiments, the media were enriched to 5% $D_2O$ thus p is 0.05. The value of N was determined by regression of the observed $M0_t$ values relative to theoretical prediction. The calculated value of N was between 21 and 26 for all sterols in all cell lines, which is consistent with previously reported values (*Lee et al., 1994*). $M0_t$ was determined by rounding N to the nearest integer. The concentrations of sterol intermediates were negligible in media and chow, except for t-MAS, which was present in chow at a concentration of 68 ng/mg.

For mouse experiments, the value of p was determined by iteratively regressing N and p values against the experimentally determined asymptotic value of $M0_t$ to minimize the root mean square value across all tissues using MIDA. The value of N for lanosterol observed in cell culture experiments (N = 21) was used as an initial value. This analysis yielded a value for p of 0.048 which was used for all tissues. The value of N for all sterols was then calculated based on the asymptotic value of $M0_t$. The value of N ranged from 20 to 28 in all tissues.

The rate of synthesis of each sterol was determined by assuming first-order kinetics, using the equation:

$$g = g_\infty \left( 1 - e^{-kt} \right),$$

where $g_\infty$ is the asymptotic value of g, t is time, and k is the rate constant. The value of $g_\infty$ and k were calculated using the Matlab Optimization Toolbox (Natick, MA). The rate of synthesis of each sterol was determined by multiplying the rate constant (k) by the concentration.

## Real-time PCR

Total RNA from tissues and cells was isolated using commercial reagents (RNA-STAT 60). cDNA was synthesized from 2 µg RNA using Taqman (Applied Biosystems, Grand Island, New York.) with random hexamer primers and amplified by PCR in 2× SYBR Master Mix (Applied Biosystems). The specific oligonucleotides for each transcript are shown in *Supplementary file 3*. The levels of each mRNA were normalized to the level of 36B4.

## Immunoblot analysis

Samples for immublotting were placed in sample buffer (10 mM HEPES, 1.5 mM $MgCl_2$, 10 mM KCl, 5 mM EDTA, 5 mM EGTA, 250 mM sucrose, pH 7.6) and homogenized using an Ultra-TURRAX homogenizer and then passed through a 22 G needle 35 times. Membranes were isolated from the homogenates by sequential centrifugation (7 min at 2200 rpm, followed by a second centrifugation at 14,000 rpm for 60 min). The pellet from the second centrifugation step was re-suspended in lysis buffer (10 mM tris–HCl, 100 mM NaCl, 1% SDS, 1 mM EDTA and EGTA, pH 6.8) and shaken for 1 hr at 4°C.

After an hour, an equal volume of solubilization buffer (62.5 mM Tris–HCl, 15% SDS, 8 M urea, 10% glycerol, 100 mM DTT, pH 6.8) and 5× loading buffer was added (Fermantas). Samples were size fractionated on 8% SDS-polyacrylamide gels and transferred to nitrocellulose (GE Healthcare) before incubating overnight at 4°C with a rabbit anti-mouse DHCR24 antibody (1:1000; Cell Signaling Technologies) in PBST (Sigma) plus 5% (wt/vol) fat free milk. Anti-mouse calnexin (1:2000; MBD International) was used to detect calnexin as a loading control. After incubation, the filter was washed and HRP-conjugated anti-rabbit IgG (1:5000; GE Healthcare) was used as a secondary antibody. The filter was scanned using a LiCor Odyssey and visualized using iS ImageStudio software.

### Transfection

Mouse DHCR24 was cloned into a PCDNA 3.1 TOPO TA vector using a commercial kit (Invitrogen). The insert sequence was verified by sequencing and the plasmid was amplified with a Maxiprep kit (Origene). The plasmid was prepared in a 1:3 ratio with Fugene 6 (Promega) (wt/vol) and diluted in Opti-Mem medium (Life Technologies). Cells were transfected by exchanging the medium for Opti-Mem medium containing 8 µg plasmid DNA at a 1:3 ratio with Fugene 6 (wt/vol). After 20 hr the medium was changed to Opti–Mem supplemented with 5% $D_2O$.

## Acknowledgements

We wish to thank David Russell, Elizabeth Parks, and Russell Debose-Boyd for helpful discussions and Zifen Wang for technical assistance. We also thank Dr Donald Glass (UT Southwestern) for assistance in isolating the skin samples used in these studies.

## Additional information

### Competing interests

HHH: Reviewing editor, *eLife.* The other authors declare that no competing interests exist.

### Funding

| Funder | Grant reference | Author |
| --- | --- | --- |
| Howard Hughes Medical Institute (HHMI) | | Helen H Hobbs |
| National Institutes of Health (NIH) | R37HL72304 | Matthew A Mitsche, Jeffrey G McDonald, Helen H Hobbs, Jonathan C Cohen |
| National Institutes of Health (NIH) | PO1 HL20948 | Matthew A Mitsche, Jeffrey G McDonald, Helen H Hobbs, Jonathan C Cohen |
| National Institutes of Health (NIH) | K01GM109317 | Matthew A Mitsche |
| National Institutes of Health (NIH) | UL1TR001105 | Helen H Hobbs |
| Clayton Foundation for Research | | Jeffrey G McDonald |

The funders had no role in study design, data collection and interpretation, or the decision to submit the work for publication.

### Author contributions

MAM, Conception and design, Acquisition of data, Analysis and interpretation of data, Drafting or revising the article; JGM, Acquisition of data, Analysis and interpretation of data, Drafting or revising the article; HHH, JCC, Conception and design, Analysis and interpretation of data, Drafting or revising the article

### Ethics

Animal experimentation: This study protocol was approved by the institutional animal care and use committee (IACUC) of the University of Texas Southwestern Medical Center (APN 2008-0321).

All studies were performed in accordance with the recommendations in the Guide for the Care and Use of Laboratory Animals of the National Institutes of Health.

## Additional files

### Supplementary files

• Supplementary file 1.  Rate constants (k) and concentrations of cholesterol biosynthetic intermediates in mouse tissues.

• Supplementary file 2.  LC-MS/MS characteristics of sterols.

• Supplementary file 3.  Primer sequences for real-time PCR of genes in the cholesterol biosynthetic pathway.

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
