## [Decision Letter]

Thank you for submitting your work entitled “Flux analysis of cholesterol biosynthesis in vivo reveals multiple tissue and cell-type specific pathways” for peer review at *eLife*. Your submission has been favorably evaluated by Randy Schekman (Senior editor) and three reviewers, one of whom is a member of our Board of Reviewing Editors.

The reviewers have discussed the reviews with one another and the Reviewing editor has drafted this decision to help you prepare a revised submission.

The following individuals responsible for the peer review of your submission have agreed to reveal their identity: Stephen Young (Reviewing editor); Joanne Kelleher and Marc Hellerstein (peer reviewers).

All three reviewers were enthusiastic about your paper and all three thought that a revised manuscript that attended to the reviewers' criticisms would be suitable for publication in *eLife*.

In their manuscript, the authors have analyzed cholesterol synthesis through the Bloch and Kandutsch-Russell pathways in different tissues and cell types. They found that no tissue used the canonical K-R pathway, but that several used a modified K-R pathway. They found that the use of the Bloch and MK-R pathways is highly variable, tissue-specific, and flux dependent. The factors that regulated use of the different pathways was not clearly delineated, but the manuscript represents an important contribution to cholesterol biochemistry.

None of the three reviewers requested additional experiments, but all three reviewers thought that it was essential for the authors to defend some of their conclusions and expand upon their interpretations of the data. Also, reviewers 2 and 3 thought that the authors needed to clarify methodological issues and in some cases qualify their conclusions. For example, the authors need to discuss how subpools of intermediates might affect their interpretations and conclusions. The authors are strongly encouraged to respond to all of the reviewers' criticisms regarding methodology and data interpretation in a revised manuscript.

Reviewer #1:

This is a beautifully prepared manuscript with great figures and novel data. The authors have worked define the importance of two different cholesterol biosynthetic pathways in different cell lines and in different mouse tissues. They have also examined the two biosynthetic pathways in response to perturbations that alter cholesterol synthetic rates. They have identified several important novel twists and turns in the pathway. The authors clearly have carved out a fruitful and intriguing topic for future research.

I have several comments. First, the authors should defend their contention that the identification of the hybrid cholesterol synthesis pathway is novel. This reviewer noted that this hybrid pathway appears to be described in the cholesterol synthesis discussion in the Wikipedia page on Smith Lemli Opitz syndrome. Of course, Wikipedia has plenty of errors. Nevertheless, I would request that the authors defend their contention that the “pathway switchover” is novel – or whether their hybrid pathway had been conceptualized and proposed previously.

Second, I would ask the authors to elaborate on the fact that plasma and liver have similar proportions of deuterium enrichment in cholesterol and precursor sterols. This is a nice finding. This reviewer believes that some sort of model would be useful to reconcile this finding with the observation that most of the cholesterol is made outside of the liver. How do the authors propose that this occurs? Please elaborate.

Also, the authors showed that expression of DHCR24 affects the pathway, but that the extent to which pathway usage depends on DHCR24 expression was not 100% consistent. One possibility is that pathway utilization depends on the concentrations, or relative concentrations, of pathway intermediates, perhaps including pre-squalene intermediates. Was there evidence for this? Can one perturb pathway utilization with low doses of squalene synthase inhibitors? Please elaborate on your thoughts about what is controlling pathway utilization.

It would be interesting, if possible, to tie this work to disease. For example, it might be interesting to compare pathway usage in different tissues of Smith Lemli Opitz syndrome mice. In those mice, is the utilization of different biosynthetic pathways in different tissues perturbed or maintained? Could you quantify the amount of 7-Dehydrocholesterol that is shunted away into oxidized derivatives? Does different pathway utilization in different tissues explain the susceptibility of different tissues to disease? Such studies are almost certainly too much for this paper, but the authors should at least put forward a more detailed description of what they consider to be the most important biochemical factors that govern pathway utilization of the two different pathways in different cell lines and tissues.

Reviewer #2:

This is an original and very well-conceived study of metabolic regulation, addressing the pathway of cholesterol biosynthesis. The results are consistent and convincing, indicating that the pathway for cholesterol synthesis in most mouse tissues – specifically, the step at which cholesterol-24,25 reduction occurs, catalyzed by DHCR24 – is late (classic Bloch pathway) or is mid-pathway, after zymosterol (modified Kandutsch-Russell [K-R] pathway). Accordingly, it is somewhat misleading to talk of an eponymous, pure K-R pathway. Variability among tissues is not explained by gene expression (mRNA levels, even for DHCR24) and may have regulatory consequences.

The methods appear solid, combining heavy water labeling and LC/MS mass isotopomer analysis with pool size measurements to measure synthesis rates and using established modifiers of cholesterol flux (cholesterol depleted media, DHCR24 overexpression). The paper is well written.

I believe that this is an unusually compelling contribution to the classic field of cholesterol biosynthesis.

There are a few important methodologic issues to clarify:

1) How much of the differences in total synthesis rates are due to the kinetic measurements (turnover rates) vs. the tissue concentrations of each metabolite? The k values are not shown (e.g., in an appendix). Would their findings have been discernable from simple measurements of metabolite concentrations (e.g., classic “cross-over plots”)?

2) The inability to detect some key biosynthetic intermediates analytically which happen to be right about where the DHCR24 crossover often occured – zymostenol, lathosterol, dehydrolathosterol – is a problem. The authors are forced to place this key step somewhere at or after zymosterol. This uncertainty, which is entirely analytic in nature, might be emphasized a bit more clearly in the text. Their one non-heavy water labeling study, with d_6_-lanosterol and d_5_-zymostenol, overcame the detectability problem and gave more detail on the cross-over site. The latter results support their general conclusion – i.e., cross-over seemingly around the dehydrolathosterol-lathosterol step, or maybe a step earlier. This supportive labeling result is worth noting in context of the greyed out parts of Figure 1.

3) The kinetic method is not accurately described as ISA, which solves for p and synthesis fraction (g) by best fit to a family of curves. The authors know p here independently (tissue water enrichment). Their equation for g (in the last paragraph of the subsection headed “Data interpretation”) is basic mass isotopomer analysis, identical for MIDA (JCI 1991, AJP 1992) and ISA (AJP 1992).

4) A point worth mentioning about Figure 2 (and subsequent figures): it is not possible to have a higher absolute flux rate in the final product of a linear pathway, in this case 7-dehydrocholesterol, than in preceding intermediates. Like water through a hose, flow cannot be greater over one length than over an earlier length. So these figures by themselves must mean a crossover into “K-R” has occurred – i.e., that K-R is not a linear pathway.

Additional data files and statistical comments:

Appendix showing kinetic rate constants (k) vs. metabolite concentrations, in their calculations of absolute synthesis rates (= k x concentration).

Reviewer #3:

This manuscript contributes to understanding the pathway for cholesterol synthesis. Specifically it demonstrates that a kinetic labeling approach using ISA may be used to resolve the flux through the Bloch vs the K-R pathway for cholesterol synthesis. A possible difficulty with this approach would be the presence of two pools of an intermediate in the pathway. Consider the possibility of a small pool involved in the biosynthetic pathway which labeled fully and quickly and a second larger pool sequestered from the biosynthesis pathway which did not turn over. Such a pool might reduce to negligible values the labeling of the total intermediate isolated from the cells and falsely give the impression that there was no flux through this poorly labeled intermediate. Clearly this does not occur for the well labeled intermediates which label to 100% with expected kinetics. However, it might be useful if the authors commented on this possibility and any experiments that might rule this out.

---

## [Author Response]

Reviewer #1:

*I have several comments. First, the authors should defend their contention that the identification of the hybrid cholesterol synthesis pathway is novel. This reviewer noted that this hybrid pathway appears to be described in the cholesterol synthesis discussion in the Wikipedia page on Smith Lemli Opitz syndrome. Of course, Wikipedia has plenty of errors. Nevertheless, I would request that the authors defend their contention that the* “*pathway switchover*” *is novel – or whether their hybrid pathway had been conceptualized and proposed previously*.

We are not aware of any previous demonstration of the hybrid pathway in a living cell or animal. The comment in Wikipedia appears to have its origin in a paper by Bae and Paik (Bae and Paik, Biochem J 326:609, 1997), who reported that zymosterol was preferred over lanosterol as a substrate for C-24 reduction in rat liver microsomes. Those authors hypothesized that the physiological pathway for conversion of lanosterol to cholesterol did not correspond to either the Bloch or the KR pathways, but rather to a reaction sequence in which C24-reduction occurred after demethylation but before the final Δ^5^-dehydrogenation and Δ^7^-reduction of the sterol nucleus. This proposal was not assessed in vivo and did not gain wide acceptance. Subsequently, Gaylor et al. noted the enzymes in the pathway act on sterols with both unsaturated and saturated side chains and so crossover between the Bloch and Kandutsch-Russell pathways could potentially occur at any point in the pathway (Gaylor J, BBRC 295:1139, 2002). The interpretation was challenged by Brown and his colleagues (Zerenturk et al. Prog Lipid Res 52:666, 2013) since desmosterol, not zymosterol, accumulates in DHCR24 deficiency. In response to this comment, we have revised the manuscript as follows:

In the first paragraph of the Discussion we have changed the statement “A previously unrecognized hybrid pathway that we have named the modified Kandutsch-Russell pathway (MK-R) exists in skin, preputial glands and brain (Figure 8, red arrows)” to: “A hybrid pathway that we have named the modified Kandutsch-Russell pathway (MK-R) exists in skin, preputial glands and brain (Figure 8, red arrows).”

We have added the following paragraph to the Discussion: “The MK-R pathway is consistent with […] a major physiological substrate for the enzyme.”

*Second, I would ask the authors to elaborate on the fact that plasma and liver have similar proportions of deuterium enrichment in cholesterol and precursor sterols. This is a nice finding. This reviewer believes that some sort of model would be useful to reconcile this finding with the observation that most of the cholesterol is made outside of the liver. How do the authors propose that this occurs? Please elaborate*.

The reviewer raises a very interesting point. If circulating sterols are largely derived from liver how are sterols synthesized in peripheral tissues transported for excretion in liver or gut? This question touches on one of the major unresolved questions in cholesterol metabolism (viz. reverse cholesterol transport), and we agree that it deserves comment. Since the pathway appears not to be captured by our present data, any model we propose would be entirely speculative. We have added the following to the Discussion: “If extrahepatic tissues contribute little to the plasma sterol […] plasma elements to the centripetal transport of peripheral tissue cholesterol.”

*Also, the authors showed that expression of DHCR24 affects the pathway, but that the extent to which pathway usage depends on DHCR24 expression was not 100% consistent. One possibility is that pathway utilization depends on the concentrations, or relative concentrations, of pathway intermediates, perhaps including pre-squalene intermediates. Was there evidence for this? Can one perturb pathway utilization with low doses of squalene synthase inhibitors? Please elaborate on your thoughts about what is controlling pathway utilization*.

We did not see an obvious relationship between concentrations of sterol intermediates and relative pathway utilization. We have added this data to the paper as requested by Reviewer 2 (see new Figure 5–figure supplement 1). We did not test squalene synthase inhibitors. Nonsterol intermediates are not detected in our assay, so we cannot exclude the hypothesis that the differences observed are due to concentrations of nonsterol these intermediates in the pathway.

We have expanded our discussion on possible factors that control pathway utilization as follows: “The relative utilization of the Bloch and MK-R pathways in cultured cells […] route of flux through the cholesterol biosynthetic pathways”.

*It would be interesting, if possible, to tie this work to disease. For example, it might be interesting to compare pathway usage in different tissues of Smith Lemli Opitz syndrome mice. In those mice, is the utilization of different biosynthetic pathways in different tissues perturbed or maintained? Could you quantify the amount of 7-Dehydrocholesterol that is shunted away into oxidized derivatives? Does different pathway utilization in different tissues explain the susceptibility of different tissues to disease? Such studies are almost certainly too much for this paper, but the authors should at least put forward a more detailed description of what they consider to be the most important biochemical factors that govern pathway utilization of the two different pathways in different cell lines and tissues*.

We agree that a disease connection would be interesting, and we looked at pathway utilization in primary fibroblasts cultured from patients with Smith-Lemli-Opitz Syndrome. In these experiments, sterol biosynthesis was reduced by about 50%, which is consistent with previous studies by Tint and Salen, and the modified pathway was used exclusively, which is consistent with the biochemical defect (the cells cannot synthesize desmosterol). Accordingly we elected not to include these data.

It is likely that the amount of 7-dehydrocholesterol that is diverted to oxidized derivatives could be determined using the methodology applied here, but it would require nontrivial method development. The assay of oxysterols requires completely different sample preparation, LC conditions (gradient), and mass spectrometry (ion source) from those used in this study.

Reviewer #2:

*There are a few important methodologic issues to clarify*:

*1) How much of the differences in total synthesis rates are due to the kinetic measurements (turnover rates) vs. the tissue concentrations of each metabolite? The k values are not shown (e.g., in an appendix). Would their findings have been discernable from simple measurements of metabolite concentrations (e.g., classic* “*cross-over plots*”*)?*

We have now included the k-values and sterol concentrations for all mouse tissues in Figure 5–figure supplement 1. We could not predict the relative flux from the substrate concentrations.

*2) The inability to detect some key biosynthetic intermediates analytically which happen to be right about where the DHCR24 crossover often occured – zymostenol, lathosterol, dehydrolathosterol – is a problem. The authors are forced to place this key step somewhere at or after zymosterol. This uncertainty, which is entirely analytic in nature, might be emphasized a bit more clearly in the text. Their one non-heavy water labeling study, with d*_*6*_*-lanosterol and d*_*5*_*-zymostenol, overcame the detectability problem and gave more detail on the cross-over site. The latter results support their general conclusion – i.e., cross-over seemingly around the dehydrolathosterol-lathosterol step, or maybe a step earlier. This supportive labeling result is worth noting in context of the greyed out parts of*
Figure 1.

To emphasize this point in the text, we have added the following statement to the Discussion: “Whereas we were not able to localize the specific DHCR24 substrate in the D_2_O labeling studies, experiments using deuterium-labeled lanosterol and zymosterol confirmed that zymosterol is the first sterol in the pathway to undergo appreciable side-chain reduction, at least in these cells.”

*3) The kinetic method is not accurately described as ISA, which solves for p and synthesis fraction (g) by best fit to a family of curves. The authors know p here independently (tissue water enrichment). Their equation for g (in the last paragraph of the subsection headed “Data interpretation”) is basic mass isotopomer analysis, identical for MIDA (JCI 1991, AJP 1992) and ISA (AJP 1992)*.

To avoid mischaracterizing our analysis we have changed “ISA” to “isotopomer analysis (IA)” and cite both MIDA and ISA.

*4) A point worth mentioning about*
Figure 2
*(and subsequent figures): it is not possible to have a higher absolute flux rate in the final product of a linear pathway, in this case 7-dehydrocholesterol, than in preceding intermediates. Like water through a hose, flow cannot be greater over one length than over an earlier length. So these figures by themselves must mean a crossover into* “*K-R*” *has occurred – i.e., that K-R is not a linear pathway*.

We agree.

Additional data files and statistical comments:

Appendix showing kinetic rate constants (k) vs. metabolite concentrations, in their calculations of absolute synthesis rates (= k x concentration).

We have added the k values as requested. Figure 5–figure supplement 1.

Reviewer #3:

*This manuscript contributes to understanding the pathway for cholesterol synthesis. Specifically it demonstrates that a kinetic labeling approach using ISA may be used to resolve the flux through the Bloch versus the K-R pathway for cholesterol synthesis. A possible difficulty with this approach would be the presence of two pools of an intermediate in the pathway. Consider the possibility of a small pool involved in the biosynthetic pathway which labeled fully and quickly and a second larger pool sequestered from the biosynthesis pathway which did not turn over. Such a pool might reduce to negligible values the labeling of the total intermediate isolated from the cells and falsely give the impression that there was no flux through this poorly labeled intermediate. Clearly this does not occur for the well labeled intermediates which label to 100% with expected kinetics. However, it might be useful if the authors commented on this possibility and any experiments that might rule this out*.

We agree that this is an important caveat and we thank the reviewer for bringing it to our attention. We have added the following caveat to the Discussion: “The approach used in this study could potentially be confounded by the presence of two pools of an intermediate in the pathway that turn over at different rates. This problem would be particularly acute if a small pool of intermediate turns over rapidly while a second, larger pool sequestered from the biosynthesis pathway turned over slowly. Under these conditions the larger pool may significantly reduce the labeling of the total intermediate isolated from the cells resulting in falsely low estimates of flux. The observation that multiple independent intermediates showed essentially identical kinetics argues against this possibility, but we cannot formally exclude it.”